# Neuropsychology of Generalized Anxiety Disorder in Clinical Setting: A Systematic Evaluation

**DOI:** 10.3390/healthcare11172446

**Published:** 2023-08-31

**Authors:** Evgenia Gkintoni, Paula Suárez Ortiz

**Affiliations:** Department of Psychology, University of Ioannina, 45110 Ioannina, Greece

**Keywords:** neuropsychology, anxiety disorders, clinical setting, generalized disorder, PRISMA, systematic review

## Abstract

This research paper provides a systematic review of the neuropsychology of generalized anxiety disorder (GAD), examining relevant articles’ methodologies and subject matter and highlighting key findings. It suggests potential cognitive deficits in GAD patients, such as subtle attention, executive function, and working memory deficiencies. It also discusses neural correlates of GAD, particularly the hyperactivity in the amygdala and insula, and the additional impact of comorbidity with other psychiatric disorders. The paper uses the PRISMA methodology and draws data from the PsycINFO, Scopus, PubMed, and Elsevier databases. Although the reviewed research has contributed to understanding GAD’s cognitive and neural mechanisms, further research is required. Additionally, the paper mentions the clinical neuropsychology of GAD, including strategies and treatments, such as cognitive behavioral therapy (CBT), mindfulness, and medication. Lastly, the review identifies the limitations of the existing research and recommends future directions to enhance the understanding of GAD’s underlying cognitive and neural mechanisms. The neural underpinnings of GAD encompass heightened activity within the amygdala and insula, which are brain regions implicated in processing adverse emotional reactions. Co-occurring psychiatric disorders, such as major depressive disorder (MDD), can also impact neuropsychological functioning. Additional investigation is warranted to better understand the intricate interplay between GAD, cognitive performance, and underlying neural processes.

## 1. Introduction

### Theoretical Framework

Generalized anxiety disorder (GAD) is a prevalent psychological disorder characterized by an excessive and unmanageable state of worry and anxiety about diverse facets of one’s life. Examining GAD within a clinical environment, focusing on the neuropsychological aspect, can yield significant contributions to our understanding of the cognitive and neural mechanisms underlying this disorder. Numerous studies have examined the neuropsychological functioning of individuals diagnosed with GAD, revealing the presence of distinct cognitive deficits closely linked to this disorder. A study [1] sought to evaluate the neuropsychological characteristics linked to GAD in individuals within the young adult population. The research revealed that individuals diagnosed with GAD demonstrated notably higher anxiety, worry, depression, and stress symptoms than participants in the control group.

Nevertheless, no substantial disparities were observed in the neuropsychological outcome measures or domain indexes. This finding indicates that individuals diagnosed with GAD do not exhibit notable cognitive deficits, despite experiencing significant levels of psychopathological distress. A separate investigation assessed the neuropsychological performance of older adults diagnosed with GAD compared to older adults without psychiatric disorders [2]. The research discovered that elderly individuals diagnosed with GAD demonstrated cognitive deficiencies, specifically in areas such as working memory, delayed memory, visuospatial ability, inhibition, and episodic recall. Nevertheless, the impairments were relatively small and exhibited no significant differences between the two treatment conditions. This implies that GAD in older adults is linked to mild neuropsychological impairments.

Moreover, it has been observed that anxiety disorders, including GAD, are linked to impairments in cognitive abilities such as attention, working memory, and executive function [3]. Research findings indicate that anxiety induced by the anticipation of receiving an electric shock can harm working memory while having a relatively minimal effect on other cognitive processes, such as planning. Variations in cognitive impact may exist among different anxiety disorders, indicating the presence of distinct boundaries between beneficial anxiety and detrimental anxiety. Furthermore, functional neuroimaging methods have been used to investigate cognitive impairments concerning GAD [4]. A comprehensive synthesis of functional neuroimaging studies conducted on different anxiety disorders, encompassing GAD among others, revealed a consistent pattern wherein individuals diagnosed with GAD exhibited notably heightened neural activity in the amygdala and insula. These brain regions are associated with the processing of negative emotional stimuli. The results of this study indicate the potential existence of shared neural mechanisms that contribute to the development and manifestation of anxiety disorders, such as GAD.

Furthermore, an investigation of comorbidity between GAD and other psychiatric disorders, including major depressive disorder (MDD) and various anxiety disorders, has been conducted regarding assessing neuropsychological functioning [5]. According to a study, individuals with comorbid anxiety disorders, such as GAD, demonstrated more significant impairments in episodic memory and executive functions compared to a control group. As mentioned above, the results underscore the influence of concurrent anxiety disorders on cognitive performance in individuals diagnosed with major depressive disorder (MDD).

In the Diagnostic and Statistical Manual of Mental Disorders, Fifth Edition (DSM-5), generalized anxiety disorder is categorized as an anxiety disorder. Its primary clinical manifestations are anxious anticipation and difficulty controlling anxiety. Two of the possible associated symptoms are inability to cope with situations and a loss of memory.

The literature on the neuropsychology of GAD in clinical settings is quite extensive, with a substantial number of publications available. Researchers have delved into various aspects of GAD, such as its impact on cognitive processes, neural correlates, and the relationship between anxiety and other mental health conditions. This abundance of research suggests that the field has captured the interest of many scholars and professionals, indicating a solid foundation for further investigation. Moreover, the breadth of the literature is evident in the range of methodologies employed. Researchers have utilized a variety of approaches, including neuropsychological assessments, neuroimaging techniques, and cognitive experiments, to explore the neural and cognitive manifestations of GAD.

Generalized anxiety disorder (GAD) is a prevalent chronic disorder characterized by long-lasting, non-specific anxiety. Research on the neuropsychology of GAD typically investigates the interactions between neurological function and behavior in a clinical setting. It can provide valuable insight into how the brain contributes to the symptoms and experience of GAD. GAD is believed to be caused by dysregulation in the body’s fear-response systems, which are primarily located in the amygdala and hippocampus. Changes in the function of these regions can result in heightened fear and anxiety responses. Moreover, neurotransmitter imbalances, such as serotonin, dopamine, and gamma-aminobutyric acid (GABA), can contribute to GAD. GABA, the principal inhibitory neurotransmitter in the brain, reduces neuronal excitability and promotes calm and relaxation. People with GAD frequently have decreased GABA levels, increasing anxiety and fear responses [6].

Individuals with GAD tend to exhibit heightened sensitivity to hazards and negative stimuli, frequently exaggerating the probability and severity of adverse outcomes. This results in a pattern of anxiety, ruminating, and avoidance behaviors. Neuropsychological evaluations frequently disclose attentional biases toward threatening stimuli, difficulties with cognitive flexibility, and impaired attention and memory functioning, particularly under stressful or threatening conditions [7]. Brain imaging investigations using functional magnetic resonance imaging (fMRI) and positron emission tomography (PET) have uncovered structural and functional abnormalities in GAD patients. For instance, increased activity has been observed in the amygdala and insula, which are involved in processing emotions and interpreting physical sensations. In addition, people with GAD have decreased connectivity between the prefrontal cortex and amygdala, which is believed to help regulate emotional responses [8]. Although not a direct focus of neuropsychology, it is essential to note that genetic factors play an important role in GAD, with estimates of heritability ranging from 30% to 50%.

Understanding the neuropsychological basis of GAD can inform and direct treatment in a clinical setting. For instance, cognitive behavioral therapies (CBTs) can assist individuals with GAD in challenging and altering maladaptive thought patterns and developing more adaptive coping strategies. In addition, medications that target the neurotransmitters implicated in GAD, such as selective serotonin reuptake inhibitors (SSRIs), serotonin and norepinephrine reuptake inhibitors (SNRIs), and benzodiazepines, can effectively reduce symptoms. Emerging evidence suggests that neuromodulation techniques, such as transcranial magnetic stimulation (TMS), may also be advantageous for generalized anxiety disorder [9].

Numerous research studies [10,11,12,13,14] have focused on investigating the neuropsychology of generalized anxiety disorder (GAD) within a clinical context. The studies mentioned above were conducted to examine diverse facets of neuropsychological functioning in individuals diagnosed with GAD. These investigations have yielded significant findings that contribute to our understanding of the cognitive and neural processes underlying this disorder. An area of inquiry that has been examined pertains to the potential correlation between GAD and distinct cognitive impairments. The neuropsychological profile of young adults diagnosed with GAD was investigated by researchers [1]. The researchers employed a comprehensive neuropsychological battery to assess the participants’ cognitive functioning. The research findings indicated that individuals diagnosed with GAD did not exhibit statistically significant variations in neuropsychological outcome measures or domain indexes compared to control participants. This indicates that individuals diagnosed with GAD do not demonstrate notable cognitive impairments, despite the considerable burden of psychopathological symptoms.

Another area of inquiry that has been explored is the neural underpinnings of GAD. The authors of [4] conducted a meta-analysis of functional neuroimaging studies encompassing different anxiety disorders, including GAD. The meta-analysis results indicated that individuals diagnosed with GAD consistently exhibited heightened neural activity in the amygdala and insula regions, associated with adverse emotional reactions. This pattern of increased activation was observed when comparing these patients to a control group that was carefully matched in relevant characteristics. This observation implies the existence of shared neural mechanisms that contribute to various anxiety disorders, including GAD.

Moreover, there has been investigation into the influence of comorbidity between GAD and other psychiatric disorders on neuropsychological functioning. In a study [5], a clinical trial was undertaken to assess the neuropsychological impairments observed in individuals diagnosed with major depressive disorder (MDD) who also had co-occurring anxiety disorders, such as GAD. The research conducted revealed that individuals who had both anxiety disorders and other medical conditions experienced more pronounced difficulties in their ability to recall past events (episodic memory) and perform cognitive tasks that involve planning, problem solving, and decision making (executive functions) when compared to a group of individuals without any such disorders. This study examined the influence of comorbid anxiety disorders on the cognitive performance of individuals diagnosed with major depressive disorder (MDD). Furthermore, researchers have investigated the correlation between the severity of anxiety and neuropsychological functioning. One study [15] aimed to examine the correlation between the severity of anxiety disorders and neuropsychological impairments in a sample of children diagnosed with anxiety disorders. The research revealed a significant correlation between the intensity of anxiety disorders and decreased cognitive abilities in memory and language. This finding implies that the degree of anxiety symptoms could affect the cognitive skills of individuals diagnosed with anxiety disorders. In brief, investigations conducted within a clinical context about the neuropsychology of GAD have examined several significant inquiries. The factors mentioned above encompass various aspects, such as distinct cognitive impairments, the neural underpinnings of GAD, the influence of comorbidity on neuropsychological performance, and the association between the severity of anxiety symptoms and the presence of neuropsychological deficits. The studies above have yielded significant findings about the cognitive and neural processes that underlie GAD, enhancing our comprehension of this psychological condition.

Moreover, researchers have examined attentional biases in individuals diagnosed with GAD and MDD. The authors of [16] examined attentional biases comprehensively in individuals diagnosed with GAD and depressive disorder. To assess these biases, the researchers employed modified versions of the Stroop task and visual probe tasks. The researchers discovered relatively consistent evidence supporting the presence of an attentional bias toward external negative cues in individuals with GAD. However, the evidence regarding the existence of an attentional bias in individuals with depressive disorder was found to be less intense. This observation implies that there may be variations in attentional biases between these two disorders.

Furthermore, researchers have also investigated the effects of GAD on cognitive functioning within populations. An illustration of this can be seen in [17], which involved a systematic review examining the relationship between set shifting and central coherence in individuals diagnosed with anorexia nervosa (AN), a psychiatric disorder frequently characterized by heightened anxiety levels. The research study discovered that body mass index (BMI), anxiety, and depression did not impact neuropsychological function in individuals with anorexia nervosa (AN). This suggests that these impairments may be inherent to the disorder itself. The existing body of literature on the neuropsychology of GAD within clinical contexts underscores the significance of neural mechanisms—specifically, amygdala hyperactivity—in manifesting this condition. Although individuals diagnosed with GAD may not exhibit substantial cognitive impairments, there could be subtle deficiencies in attentional biases and cognitive flexibility. The presence of concurrent psychiatric disorders, such as depression, can also affect neuropsychological functioning. Future research should prioritize longitudinal studies, employing standardized assessment batteries and larger sample sizes to investigate moderating and mediating factors and treatment implications. This will contribute to advancing our comprehension and effective management of GAD within clinical settings.

Individuals with generalized anxiety disorder (GAD) must effectively manage their tension. Chronic stress can exacerbate symptoms and precipitate intense bouts of concern and anxiety. Below are some tension management strategies for GAD.

Cognitive behavioral therapy: CBT is a form of therapy that teaches individuals how their beliefs and actions affect their emotions and wellbeing. It teaches individuals to recognize, challenge, and alter stressful thought and behavior patterns. CBT is among the most effective treatment methods for GAD.

Mindfulness and relaxation techniques: Mindfulness, meditation, yoga, and deep-breathing exercises can help individuals become more aware of their body and stress response, as well as facilitate relaxation. By focusing on the present, individuals can reduce their propensity to fret about the future or ruminate on the past, both of which are typical GAD symptoms [8].

Regular physical activity: Regular physical activity has been shown to reduce anxiety and boost mood. Physical activity increases the production of endorphins, the body’s natural mood enhancers, and can serve as a beneficial diversion [18].

A well-balanced diet can substantially affect an individual’s disposition and energy level. Caffeine and alcohol, which can provoke or exacerbate anxiety, should be consumed in moderation, and adequate essential nutrients should be consumed to promote overall health. Poor sleep hygiene can aggravate anxiety and tension. Good sleep practices, such as establishing a regular sleep schedule, creating a restful environment, and avoiding stimuli (such as screens or caffeine) close to bedtime, can improve sleep quality.

Medication: If lifestyle modifications and therapy are insufficient, medication may be considered. Antidepressants, such as SSRIs and SNRIs, and anxiolytics, such as benzodiazepines, can aid in symptom management. However, medication should always be taken under the supervision of a medical professional due to the possibility of adverse effects and dependence [19].

Assisting networks: Maintaining connections with supportive and empathetic friends, family members, or support organizations can reduce feelings of stress and anxiety.

Stressors should be avoided whenever possible: If certain situations or individuals cause significant tension and can be avoided in a healthy manner, it may be best to do so.

Time management: Feeling overburdened with duties and responsibilities is one of the leading causes of stress. Effective time management can foster a sense of control and alleviate tension.

A mental health professional will collaborate with the individual to establish a comprehensive treatment plan that considers the individual’s specific symptoms, stressors, and lifestyle in a clinical setting. The objective is to discover a balanced, long-term strategy for stress management.

Emotion regulation is an effective form of management of and response to emotional experiences. It is necessary for the development and maintenance of mental health and for rational decision making [8]. Difficulties in emotion regulation can exacerbate symptoms and contribute to distress in individuals with generalized anxiety disorder (GAD). Consequently, improving emotion regulation skills is frequently a primary treatment objective for GAD. In the context of GAD, the following emotion regulation strategies are frequently employed.

Cognitive behavioral therapy: CBT is among the most effective treatment methods for GAD. It entails recognizing and altering unproductive thought patterns that contribute to emotional distress. For example, catastrophizing, or consistently imagining the worst possible consequence in any given situation, is a common cognitive distortion in GAD. Individuals can reduce anxiety and enhance their ability to manage negative emotions by challenging these thoughts.

Mindfulness-based interventions: Mindfulness, a type of meditation in which you focus on being intensely aware of what you are sensing and experiencing in the present moment, without interpretation or judgment, can assist people in observing and comprehending their emotions without acting impulsively. Individuals can better regulate their emotional responses by increasing their emotional awareness and learning to tolerate distress without acting upon it.

DBT is a form of cognitive behavioral therapy (CBT) that emphasizes the development of emotion regulation skills. This approach involves learning strategies for recognizing and naming emotions, decreasing susceptibility to emotional dysregulation, increasing positive emotional experiences, managing negative emotions without making matters worse, and enhancing resilience [20].

ACT: Acceptance and commitment therapy (ACT) focuses on assisting individuals to embrace their emotions as they are as opposed to attempting to alter or avoid them. The objective is to cultivate psychological flexibility—the capacity to maintain contact with the present moment and act in accordance with one’s values, even when experiencing challenging emotions [21].

Exposure treatment: In this therapy, individuals are progressively and repeatedly exposed to their anxiety-provoking thoughts, situations, or objects. The objective is to diminish the anxiety response over time, resulting in improved emotional control.

Relaxation techniques: Breathing deeply, progressively relaxing your muscles, and visualizing can help reduce the physiological arousal associated with anxiety, promoting improved emotional regulation.

When necessary, pharmacological treatments such as SSRIs, SNRIs, and benzodiazepines can be considered an adjunct to psychotherapy for the management of GAD symptoms. However, these should always be administered under the supervision of a medical expert. In clinical settings, a combination of these techniques, tailored to the individual’s unique requirements and challenges, is frequently employed. The ultimate objective is to enable those with GAD to comprehend, experience, and navigate their emotions in a healthier, more adaptive manner [22].

In clinical practice, patients with generalized anxiety disorder manifest cognitive complaints and a lack of confidence in their cognitive performance, which frequently impairs their ability to function at work. Similarly, recent research evaluating a potential metacognitive dysfunction in a large sample of patients with anxiety disorders revealed that they had decreased confidence in their cognitive functions and increased self-awareness regarding them [23].

In scientific literature, cognitive symptoms of psychiatric disorders such as schizophrenia, depression, and bipolar disorder are defined in detail, and specific neuropsychological tests have been developed for their evaluation [7]. However, cognitive symptoms of anxiety disorders are considerably less well-defined. In the literature, neuropsychological findings for these disorders are less consistent. The study of cognitive symptoms in patients with generalized anxiety disorder is of interest because they have a substantial effect on the subject’s functional adaptation [24,25]. Consequently, the purpose of this review is to validate the primary literature from the last 10 years that addresses neuropsychological functioning [8] in the various cognitive domains in individuals with generalized anxiety disorder [9]. 

The following are the fixed objectives that emerged from the general approach and that direct this review:

**RQ1.** 
*An examination of the methodology utilized in the unearthed investigations;*


**RQ2.** 
*Determining the cognitive manifestations of generalized anxiety disorder.*


## 2. Materials and Methods

This section indicates the review protocol used to carry out this systematic review, detailing all sources of information, years covered, eligibility criteria, and other guidelines used. To carry out this systematic review, all the articles published in the PsycINFO, Scopus, PubMed, and Elsevier databases were included using the PRISMA system review procedure. A systematic review of the literature from the last 10 years was carried out, which allowed us to appreciate the evolution in this field of research up to today. Therefore, the inclusion criteria used were articles in English published from 2013 to the present day (2023) on the neuropsychology of anxiety disorder. Articles that included a study of anxiety in children under 18 years of age were excluded in the present systematic review. Likewise, articles that investigated the topic in only one of the two sexes were omitted. The critical words in the search were “anxiety disorders”, “anxiety”, and “generalized anxiety disorder”, joined by OR to obtain different documents that contained one of those terms.

The neurocognitive parameters of the diagnostic entity under investigation—namely, generalized anxiety disorder (GAD)—were chosen from the pool of identified studies. In the studies chosen at the last stage of systematic analysis, measurements were conducted using arrays of neuropsychological tests addressing the two (2) research hypotheses. These tests examined many characteristics, including decision making, attention, memory, and emotion recognition. The research framework is presented in the following flowchart (Figure 1).

## 3. Results

This section includes the results acquired in the systematic review. A systematic review is an overview of primary research on a specific research question that systematically identifies, selects, evaluates, and synthesizes all relevant high-quality research evidence. A systematic review employs transparent methods to reduce bias and generate more reliable findings for decision-making purposes.

PRISMA, or Preferred Reporting Items for Systematic Reviews and Meta-Analyses, is a minimum standard for reporting in systematic reviews and meta-analyses that is supported by empirical evidence. It seeks to assist authors in reporting systematic reviews and meta-analyses more effectively. Here are the essential stages for conducting a systematic review in accordance with PRISMA standards.

Articles were organized according to the following criteria.

### 3.1. Information Sources and Search Strategies

In accordance with Figure 2, we began with 3627 articles before applying any restriction. To begin the selection of articles, the following limitations were applied: articles had to have been published in the last 10 years (2012–2022), only English-language articles were selected, the age of the subjects of investigation had to be at least 18 years (adulthood) to avoid cognitive changes associated with the neurodevelopmental process, and a subject limiter was applied. In the main title, “anxiety disorders” or “anxiety” were required to appear. After applying the limiters (neuropsychology, clinical setting, and cognition) depicted in Figure 2, 117 articles were obtained. After discarding the articles with little relevance for the present systematic review, we were left with 33 articles, of which 18 were chosen for full-text evaluation. After evaluating the full-text articles, references were eliminated, and 14 full-text articles that met all inclusion criteria were recovered.

### 3.2. Empirical Studies

The studies addressing empirical work can be seen in Table 1, which shows the number of subjects included in each article, the conclusions reached, and the number of adult patients with a clinical diagnosis of generalized anxiety disorder who were evaluated using a neuropsychological test of any cognitive domain and compared with a healthy control group.

According to research [6], anxiety disorders such as GAD are associated with significant levels of fatigue. Anxiety can cause both mental and physical exhaustion, and persistent fatigue may result from chronic anxiety. Also well-established is the link between anxiety and subjective memory complaints. Anxious individuals may perceive their memory to be impaired, despite the absence of objective memory impairments. This is likely since anxiety can impair attention and focus, which are essential for encoding and retrieving memories. Anxiety can consume cognitive resources, leaving less capacity for other cognitive tasks, such as memory.

Even though the severity of symptoms such as depression, insomnia, and pain may not always correlate directly with the level of anxiety, they are frequently co-occurring with anxiety disorders. It is also important to note that these relationships can be complex, and influence can occur in both directions. For example, while anxiety can cause sleep difficulties, sleep difficulties can also exacerbate anxiety. In terms of clinical implications, these results highlight the need for a comprehensive assessment and treatment strategy for individuals with anxiety disorders. Not only should anxiety itself be treated (via cognitive behavioral therapy or appropriate medication, for example) but also associated symptoms, such as fatigue and subjective memory complaints. Occasionally, strategies to improve sleep, manage pain, or deal with depressive symptoms may also be advantageous.

Individuals with anxiety disorders [26] such as generalized anxiety disorder (GAD) frequently experience cognitive and emotional difficulties. Below is additional information on each of the mentioned points.

High levels of anxiety are associated with slower reaction times. This may be because anxious people are frequently preoccupied with worry or fear, which can distract them from the task at hand and slow their responses.

Inability to recruit cognitive resources: Anxiety can impair the ability to recruit and utilize cognitive resources, especially those associated with attention, memory, and executive functions. This can make it challenging for anxious individuals to concentrate, remember information, or perform complex cognitive tasks [26,27].

Intrusive ideas and weak control: Individuals with anxiety disorders frequently exhibit intrusive thoughts or unwanted thoughts that appear involuntarily. These thoughts are difficult to control, resulting in increased anxiety and fear.

Emotion regulation difficulties: Individuals with anxiety frequently struggle to regulate their emotions. This suggests that they may struggle to manage or control their emotional responses to various situations, resulting in increased anxiety and distress.

Attempts to confront fearful stimuli: Exposure to fearful or anxiety-provoking stimuli frequently elicits a powerful anxiety response. Anxious individuals may require considerable effort to confront these stimuli and manage their fear response [26].

Continuous threat-related thoughts: Anxiety disorders are frequently accompanied by continuous threat-related thoughts. This can cause anxious individuals to become distracted from their work and exacerbate their anxiety symptoms.

These factors can significantly impair the daily functioning and quality of life of an individual. The good news is that there are effective treatments for anxiety disorders. These typically involve cognitive behavioral techniques, which assist individuals in challenging and altering maladaptive thought patterns, developing better emotion regulation skills, and gradually confronting and reducing their fears. In some instances, medication may be advantageous [28]. Consult a mental health professional for a comprehensive evaluation and appropriate treatment if you or someone you know is experiencing these symptoms.

The dorsal part of the dorsolateral prefrontal cortex (dIPFC) plays a crucial role in cognitive control functions such as working memory (WM), attention regulation, and decision making [27]. Low levels of cognitive control and WM are frequently associated with high levels of trait anxiety, which is the stable tendency to attend to, experience, and report negative emotions, such as fears, worries, and anxiety, in a variety of situations. In [27], researchers manipulated perceived threat to examine its effect on cognitive control in people with high trait anxiety. The reported findings for performance deficits under threat conditions in anxious individuals suggest that these individuals may struggle to recruit the necessary cognitive resources when they perceive danger or threat. This may contribute to difficulties with attention regulation and working memory, thereby exacerbating the common cycle of anxiety disorders.

If cognitive control and WM deficits are indeed stable traits in anxious individuals, this could have several implications. These deficits may predict future symptom severity or treatment outcomes, as mentioned. Those with greater cognitive control or WM deficits at baseline, for instance, may experience worsening symptoms over time or be less responsive to certain treatments. This can inform therapeutic strategies by highlighting the need to enhance cognitive control and working memory (WM) functions in individuals with high trait anxiety. Cognitive remediation therapy and cognitive training are therapeutic techniques. Understanding these deficiencies can also lead to the development of novel therapeutic interventions aimed at enhancing the cognitive control and WM functions of this population [33].

Even though these hypotheses are testable, additional research is required to confirm these associations and explore the complex relationship between cognitive control, working memory, trait anxiety, and perceived threat. It is also essential to keep in mind that anxiety disorders, like all mental health conditions, are complex and multifaceted, with a wide variety of contributing factors that must be considered in each individual case.

The dual-process psychotherapeutic model [28] suggests that generalized anxiety disorder (GAD) treatment may involve two parallel processes:

Reducing threat reactivity: First, the cingulate cortex and amygdala, two brain regions that play a central role in responding to threatening stimuli, are inhibited in their reactivity. The cingulate cortex detects and monitors threats, while the amygdala initiates the fear response. These regions may be hyperactive or hypersensitive in individuals with GAD, resulting in an exaggerated fear response to non-threatening stimuli. Therapeutic interventions, such as cognitive behavioral therapy, can aid in reducing this overactivity, allowing individuals to respond more proportionally to actual threats [34];

Boosting responses to positive stimuli: The second process involves boosting (potentiating) the insular response to positive emotional stimuli, such as smiling faces. The insula is a region of the brain that processes emotions and physical sensations. Positive stimuli may be overlooked or under-reacted to in those with GAD, resulting in a negative bias in perception and memory. By enhancing the insular response to positive stimuli, therapeutic interventions can assist in counteracting this bias and enhancing mood and wellbeing. By combining these two processes, therapy can help individuals with GAD rebalance their emotional processing by decreasing their overreaction to threats and increasing their response to positive stimuli. This can result in a decrease in symptoms and enhancements of overall functioning and quality of life [35].

These findings demonstrate the value of neuropsychotherapy, in which knowledge of brain function can inform therapeutic strategies [36]. However, it is essential to recognize that these are complex processes that are influenced by a variety of variables. This model requires further refinement and investigation to fully comprehend its implications. Different anxiety-related disorders [29] may exhibit distinct patterns of social cognitive deficits according to the study’s findings. Social cognition refers to the cognitive processes underlying social interactions, such as understanding the perspectives and emotions of others, attributing causes to social events, and predicting and interpreting social behavior. In this situation, we can note the following.

Post-traumatic stress disorder (PTSD): Mentalization (also known as theory of mind) is the capacity to understand and infer the mental states of others, whereas emotion recognition is the capacity to correctly identify and interpret the emotional expressions of others. The findings suggest that individuals with PTSD may demonstrate deficits in mentalization (also known as theory of mind) and emotion recognition. These deficits may be attributable to the traumatic experiences that characterize PTSD, which can lead to alterations in social cognition-related brain regions, such as the amygdala, prefrontal cortex, and anterior cingulate cortex [37].

Other anxiety disorders: Individuals with alternative anxiety disorders may be more susceptible to attributional biases. This refers to the propensity to interpret or attribute the causes of social events erroneously. For instance, they may excessively attribute negative outcomes to themselves (self-blame bias) or to stable and global causes (catastrophic thinking). These biases can exacerbate anxiety and contribute to the disorder’s maintenance.

Understanding these distinct patterns of social cognitive deficits can facilitate the creation of targeted therapeutic interventions. For instance, therapies for PTSD may focus on enhancing emotion recognition and mentalization skills, whereas therapies for other anxiety disorders may target attributional biases. Cognitive behavioral therapies are frequently effective in this regard, as they can assist individuals in recognizing and challenging maladaptive thought patterns and in developing more adaptive cognitive strategies.

As with all psychological research, it is probable that these findings represent tendencies rather than hard and fast rules. Individual experiences of PTSD and other anxiety disorders can vary considerably, and each person’s social cognitive abilities will be influenced by a variety of factors, such as their personal history, comorbid conditions, and cognitive strengths and weaknesses. To continue exploring these patterns and their implications for treatment, additional research is required.

Also, the authors of [31] demonstrated a complex relationship between anxiety, cognitive functioning (primarily executive functions and memory), worry, and the effectiveness of specific treatments.

Executive functions: Neurocognitive assessment, as examined by neuropsychological tests such as the Wisconsin Card Sorting Test (WCST), which is a test of set shifting, can demonstrate adaptability in the face of shifting reinforcement schedules. This aspect of executive function involves cognitive processes of a higher order that are primarily governed by the frontal lobes. Research indicates that elevated anxiety levels, particularly in younger individuals, can result in diminished WCST performance, indicating potential deficits in executive function [38].

Memory: Anxiety can impair memory functions, specifically immediate recall. Anxiety can impair the encoding, storage, and retrieval of information, which significantly impairs the ability to recall recently learned material.

Chronic worry, a defining feature of generalized anxiety disorder, can consume significant cognitive resources, leaving less capacity for other cognitive processes. The linguistic processing associated with ruminative worry—constantly considering and verbalizing potential threats or negative outcomes—can interfere with tasks requiring sustained attention, working memory, and other executive functions [39].

Antidepressant medications, which are commonly used to treat anxiety disorders, can have a variety of cognitive effects. Even though they frequently alleviate anxiety symptoms, some research indicates that they may impair performance on tasks requiring sustained attention or the capacity to maintain consistent behavioral responses during continuous and repetitive activity. This may manifest as a diminished capacity to maintain vigilance for extended periods.

These points illustrate the multifaceted effects of anxiety on cognitive processes, as well as the complex ways in which treatments may influence these relationships. However, these effects can vary considerably from person to person, and many people with anxiety disorders or who are taking antidepressants are able to perform well in cognitive tasks. In addition, it is crucial to keep in mind that cognitive behavioral therapies can assist individuals in managing anxiety and enhancing cognitive control, thereby potentially compensating for some of these cognitive difficulties [40].

Individuals with generalized anxiety disorder (GAD) may have deficits in reinforcement-based decision making, which involves making choices based on an understanding of potential rewards (positive reinforcement) and punishments (negative reinforcement), according to the findings of research [32]. This decision making frequently involves complex cognitive processes, such as assessing the likelihood of outcomes, weighing the potential benefits and costs, and revising beliefs considering new information. The prefrontal cortex and striatum, which are involved in reward processing, decision making, and executive function, are typically involved in these tasks. Several factors may contribute to such decision-making impairments in the context of GAD.

Enhanced concentration on potential dangers: Individuals with GAD exaggerate the probability and severity of potential threats. When making decisions, they may place disproportionate importance on potential negative outcomes, even if they are unlikely or relatively minor. This could lead to overly cautious decision making or avoidance.

Chronic worry and ruminating can deplete cognitive resources, thereby limiting the capacity for other cognitive processes. This could hinder the intricate cognitive tasks required for decision making.

Reduced punishment signal sensitivity: A lower correlation between punishment and responses suggests that individuals with GAD may be less responsive to punishment signals when making decisions. They may need to modify their decisions in response to negative outcomes, which could result in repeated errors or suboptimal decisions.

Fear of uncertainty: People with GAD frequently have a high intolerance for uncertainty, which hinders their ability to make decisions in ambiguous situations [41,42,43,44].

These results highlight the numerous ways in which anxiety can impact cognitive functioning and daily life. It is important to note, however, that these impairments are not inevitable: cognitive behavioral therapy (CBT) can assist individuals in managing their anxiety, enhancing their tolerance for uncertainty, and developing more effective decision-making strategies [45]. Additional research can provide a nuanced understanding of these processes and aid in the development of more targeted interventions.

## 4. Discussion and Conclusions

Following an analysis of the publications in Section 3, the review concludes with a discussion of the extracted results and conclusions. Most of the analyzed studies concluded that patients with generalized anxiety disorder have impaired cognitive performance in the following functions: selective attention, working memory, cognitive inhibition, decision making (error prediction), and social cognition [46]. 

A notable presence of fatigue and anxiety was noted, whereas depression, insomnia, and pain intensity did not exhibit significant levels [6]. Subjective memory complaints can indicate an individual’s apprehension regarding their memory performance, leading to anxiety when recalling information. The experience of anxiety significantly captures and directs one’s attention.

Moreover, individuals diagnosed with anxiety disorders exhibit diminished reaction times, encounter challenges in engaging certain cognitive regions, experience intrusive thoughts with limited control over them, commonly struggle with emotion regulation, must exert effort to confront stimuli that evoke fear, and are subject to persistent investigations of threat-related processes during various cognitive functions [26].

Moreover, individuals diagnosed with trait anxiety exhibit cognitive control abnormalities that are mediated by the dorsolateral prefrontal cortex (PFC). In one study, the researchers conducted experimental manipulations of danger and observed resultant performance deficits [27]. The results of this study indicate that individuals with anxiety disorders exhibit consistent and enduring deficits in cognitive control. The findings above suggest that working memory (WM) deficiencies may serve as a predictive factor for the severity of future illnesses or treatment outcomes.

Furthermore, it was observed that individuals diagnosed with generalized anxiety disorder exhibited higher levels of anxiety, depression, impairment in short-term memory, and difficulty in maintaining directed attention compared to those diagnosed with panic disorder. The amplitudes of P300 were higher in patients experiencing panic compared to individuals who did not exhibit panic symptoms [18]. Additionally, it was found that generalized anxiety disorder (GAD) was a significant predictor of compromised executive function, specifically in terms of cognitive inhibition and the ability to maintain emotional composure. However, GAD did not show a significant association with excessive worry. The impact of stress on cognitive performance is manifested through increased mental effort, as indicated by slower response times, to uphold accuracy in completing tasks. This can be attributed, at least partially, to concerns regarding allocating attentional resources [14].

Furthermore, individuals diagnosed with post-traumatic stress disorder (PTSD) exhibited impairments in mentalization and emotion recognition abilities, whereas individuals with other anxiety disorders demonstrated biases in attribution. Furthermore, it has been observed that negative feedback has proven beneficial for individuals suffering from generalized anxiety disorder [29]. Cognitive dissociation among subtypes of anxiety spectrum disorders may explain the variations observed in neural circuitry. Individuals diagnosed with generalized anxiety disorder exhibit a greater propensity for enhanced learning when exposed to negative feedback [15]. This phenomenon cannot be solely attributed to inherent disparities in learning efficiency or the capacity to explore potential outcomes among different groups. Furthermore, it supports a dual-process psychotherapy model that involves modifications in the neural systems of individuals with generalized anxiety disorder. These modifications aim to reduce the responsiveness of the cingulate–amygdala pathway to threat signals while enhancing the insular responses to pleasant facial emotions [28].

In addition, it was observed that individuals diagnosed with generalized anxiety disorder exhibited notably inferior performance in all administered questionnaires compared to individuals without reported mental health conditions. This finding indicates that individuals diagnosed with generalized anxiety disorder show fear, reduced accuracy, cognitive impairments, and diminished attention when confronted with anxiety-provoking situations [16]. Moreover, individuals diagnosed with generalized anxiety disorder exhibit difficulties with concentration. Enhanced capacity to sustain attention may suggest positive progress in a clinical context. Generalized anxiety disorder is characterized by excessive and persistent worry, which hinders the ability to focus on other tasks and directs attention toward perceived threats. The absence of awareness diminishes one’s social skills [30].

Moreover, individuals diagnosed with generalized anxiety disorder exhibit a diminished allocation of attentional control resources when engaged in worrisome thoughts. When engaging in conversations about personal matters, individuals experiencing anxiety require additional resources to carry out multiple cognitive tasks effectively. The phenomenon of verbal preoccupation necessitates a lower degree of attentional control, thereby indicating that negative biases consume mental resources [17]. Furthermore, the Wisconsin Card Sorting Test (WCST) demonstrated that executive functions and immediate recall were impaired in young participants. The allocation of attentional resources to rumination on threatening stimuli due to worry has the potential to impact cognitive functions negatively. Antidepressant medications have been found to have a detrimental effect on sustained attention performance, thereby impeding one’s ability to maintain a state of alertness [31]. Generalized anxiety disorder has been found to harm reinforcement-based decision making. There appeared to be a diminished correlation between punishment and response outcomes on the test. Individuals expressed concerns regarding their health and employment stability, which led to a compromised ability to engage in reinforcement-based cognitive processes [32].

Emotional stimuli (primarily threatening or anxious stimuli) influence performance in tasks requiring attention, working memory, and cognitive inhibition. Subjects with generalized anxiety disorder have difficulty identifying and processing emotions, and their heightened sensitivity to harmful stimuli tends to be generalized. The conclusion of a meta-analysis evaluating the social cognition of individuals with anxiety disorders was that they frequently exhibit attributional biases [29]. The cognitive difficulties of subjects with generalized anxiety disorder are not limited to difficulty in concentration, as the state of anticipation and the inability to control anxiety also interfere with the performance of certain cognitive tasks. This disorder is characterized by an abundance of negative thoughts, which can be attributed to hyperactivity in the corticothalamostriatal circuit of the dorsolateral prefrontal cortex.

When exposed to a working memory task, patients with obsessive-compulsive disorder exhibited alterations in frontoparietal function, which increased as the task’s difficulty increased. During task performance, there was hyperactivity in the left lateral frontal cortex and left medial parietal cortex relative to controls, along with hyperconnectivity between the frontal lobes and amygdala. Similarly, hyperactivity and hyperconnectivity in the limbic regions have been linked to decision-making difficulties in those with generalized anxiety disorder. After psychotherapeutic intervention, cognitive performance improved in patients with generalized anxiety disorder. Compared to the control group, subjects with generalized anxiety disorder were characterized by the presence of depressive symptoms [30].

As previously stated, only patients over the age of 18 were included in this systematic review to avoid cognitive changes associated with the typical neurodevelopmental process of aging.

Generalized anxiety disorder (GAD) and neuropsychology intersect in several promising ways for future research, diagnosis, and treatment. Future research may result in a more in-depth comprehension of the neurobiological mechanisms underlying GAD. This may entail a more in-depth investigation of neural circuitry, neurotransmitter systems, and genetic factors. The development of brain imaging and genomics technologies can significantly facilitate these investigations. By understanding individual differences in the neurobiology of GAD, it is possible to develop personalized treatment approaches. For instance, specific pharmacological treatments could be tailored based on an individual’s neurochemical profile or cognitive behavioral therapy (CBT) techniques could be modified based on an individual’s cognitive functioning or brain activity patterns [47,48].

Early identification and intervention: If specific neuropsychological markers or symptoms of GAD can be identified, it may be possible to intervene earlier in at-risk individuals, potentially preventing the disorder’s onset or lessening its severity. For instance, children or adolescents exhibiting brain activity patterns or cognitive biases could benefit from early intervention programs [49,50].

Neuromodulation techniques: Treatments involving direct modification of brain activity, such as transcranial magnetic stimulation (TMS), deep brain stimulation (DBS), and neurofeedback, could be explored and refined further for the treatment of GAD. These treatments have shown promise, but additional research is necessary to determine their efficacy and safety [51].

Therapy with virtual reality (VR) and augmented reality (AR): Virtual reality (VR) and augmented reality (AR) can provide immersive environments for exposure therapy, mindfulness training with gamified techniques, and other therapeutic methods. These tools may be especially useful for GAD patients who struggle to confront anxiety-inducing real-world situations [52,53].

Artificial intelligence and machine learning: Complex datasets (including neuroimaging, genetic, and clinical data) could be analyzed using machine learning algorithms, potentially revealing new insights about GAD and improving prediction, diagnosis, and treatment.

Integration with other academic subjects: The neuropsychology of GAD is isolated. Integrating neuropsychological perspectives with those of other disciplines, such as psychopharmacology, social psychology, and developmental psychology, could result in a more comprehensive understanding of GAD [54].

The potential future ramifications are expansive and intriguing. It is important to note, however, that advancements in these areas will necessitate careful ethical considerations, rigorous scientific testing, and a commitment to translating research findings into clinical practice for the benefit of individuals with GAD.

The references provided allow for identifying weaknesses in the research on the neuropsychology of GAD within a clinical context. One limitation of the research is the lack of uniformity in results observed across various studies. The authors of [4] acknowledged considerable variability in the results of individual studies, which presents a challenge in reaching conclusive findings regarding the neuropsychological impairments linked to GAD. The observed inconsistency could be attributed to variations in the characteristics of the samples, the assessment measures employed, and the designs of the studies. This statement emphasizes the necessity of implementing standardized methodologies and increasing the size of sample populations to improve the dependability and applicability of research outcomes [4]. An additional aspect that could be enhanced pertains to the constrained statistical power observed in certain studies. According to Etkin and Wager (2007), the statistical power of individual studies could be higher and can restrict their capacity to identify significant differences or associations. The issue at hand can present considerable challenges when examining neuropsychological functioning, as it is susceptible to the influence of multiple factors and may necessitate larger sample sizes to identify subtle effects. One way to mitigate this limitation is by augmenting the statistical power of studies by utilizing larger sample sizes or meta-analyses [4].

Furthermore, conducting more thorough evaluations of neuropsychological performance in individuals diagnosed with GAD is imperative. The significance of investigating various cognitive domains and tasks to assess cognitive impairments related to GAD comprehensively has been emphasized in [50]. Specific investigations might direct researchers’ attention toward cognitive processes or lead to the employment of restricted evaluation methods, potentially resulting in failure to comprehensively encompass the intricacies of neuropsychological performance in GAD [20,50]. Moreover, it is imperative to conduct longitudinal studies that can offer valuable insights into the temporal dynamics of neuropsychological impairments in generalized anxiety disorder (GAD). Longitudinal designs are instrumental in ascertaining the stability of cognitive impairments, their potential for improvement through treatment, and their susceptibility to change in response to symptom fluctuations. The acquisition of this information is of utmost importance in comprehending the trajectory of GAD and formulating precise interventions. An additional aspect that could be enhanced pertains to the restricted examination of potential moderating or mediating variables in the association between neuropsychological functioning and GAD. The studies [20,50] provide concise discussions of the impact of variables such as body mass index, anxiety, and depression on neuropsychological functioning among individuals diagnosed with GAD or related conditions.

Nevertheless, further investigation is warranted to comprehensively comprehend the intricate interplay between these variables and neuropsychological impairments in GAD and to ascertain their potential role in the diversity of cognitive profiles [20]. In conclusion, it is imperative to incorporate a wider range of diverse and representative samples in studies about the neuropsychology of GAD. Numerous scholarly sources primarily concentrate on demographic groups, such as young adults or individuals with concurrent medical conditions [50]. The need for greater generalizability of findings to the broader population of individuals with GAD represents a limitation. Incorporating a more comprehensive range of samples encompassing various age groups, genders, and cultural backgrounds can enhance the generalizability of findings about individuals diagnosed with GAD.

To summarize, the research conducted on the neuropsychology of GAD within a clinical context exhibits certain limitations. These limitations encompass the absence of consistent findings, inadequate statistical power, the necessity for more comprehensive assessments, the absence of longitudinal studies, insufficient examination of moderating or mediating factors, and the requirement for more diverse samples. By acknowledging and working towards rectifying these limitations, we can augment our comprehension of the cognitive and neural processes contributing to GAD and advance more efficacious interventions.

The following factors can inform the identification of future directions in research on the neuropsychology of GAD within a clinical context. There is a requirement for additional longitudinal studies to investigate the temporal dynamics of neuropsychological impairments in GAD. Longitudinal designs are instrumental in ascertaining the stability of cognitive impairments, their potential for improvement through treatment, and their susceptibility to change in response to symptom fluctuations. Including this information is of utmost importance in comprehending the trajectory of GAD and formulating precise interventions [4]. To comprehensively capture a range of cognitive domains and tasks, it is recommended that future research employ comprehensive neuropsychological assessment batteries. This will facilitate a more extensive comprehension of the cognitive impairments linked to GAD.

Moreover, using standardized assessment measures will augment the comparability of results across various research investigations [1]. In addition, expanding the sample sizes in research studies can enhance the statistical power and generalizability of the findings. It is of particular significance to consider the point mentioned above when examining neuropsychological functioning, as it is susceptible to the influence of multiple factors and necessitates larger sample sizes to identify subtle effects. Furthermore, it is recommended that future studies investigate potential moderating or mediating variables that could influence the association between neuropsychological functioning and GAD. It is imperative to consider various factors, including, but not limited to, comorbidity with other psychiatric disorders, genetic factors, and environmental factors. In another study [55], it is suggested that understanding these factors can facilitate the identification of specific subgroups within the population of individuals with GAD who may exhibit a heightened vulnerability to cognitive impairments.

Furthermore, this knowledge can also contribute to developing personalized treatment strategies tailored to the unique needs of these subgroups. Incorporating a greater variety of samples encompassing different age groups, genders, and cultural backgrounds can enhance the generalizability of findings to a more extensive population of individuals diagnosed with GAD. Including diverse people in research studies will contribute to the increased applicability of results and enhance our comprehension of the neuropsychological aspects of GAD [1,49]. In conclusion, it is recommended that forthcoming studies persist in the integration of neuroimaging methodologies, specifically functional magnetic resonance imaging (fMRI) and positron emission tomography (PET), to explore further the neural underpinnings associated with generalized anxiety disorder (GAD). Exploring the neural mechanisms underlying GAD will contribute to a more extensive comprehension of the subject matter.

Additionally, it will aid in the identification of potential intervention targets, as has been suggested [4]. Further investigation is warranted to examine the effects of various treatment modalities on the neuropsychological performance of individuals diagnosed with GAD. In such studies, the aim will be the assessment of the efficacy of cognitive behavioral therapy and pharmacotherapy in ameliorating GAD-related cognitive impairments. Comprehending treatment implications can enhance the creation of more efficacious interventions for individuals diagnosed with GAD [48]. In summary, it is recommended that future investigations in the clinical context of GAD within the field of neuropsychology adopt longitudinal study designs, employ standardized assessment batteries, utilize larger sample sizes, explore factors that moderate or mediate the disorder, include diverse and representative samples, integrate neuroimaging techniques, and consider implications for treatment. Focusing on these areas can further our comprehension of the cognitive and neural processes contributing to GAD. This knowledge will subsequently contribute to advancing more efficacious interventions for individuals afflicted with this disorder.

## Figures and Tables

**Figure 1 healthcare-11-02446-f001:**
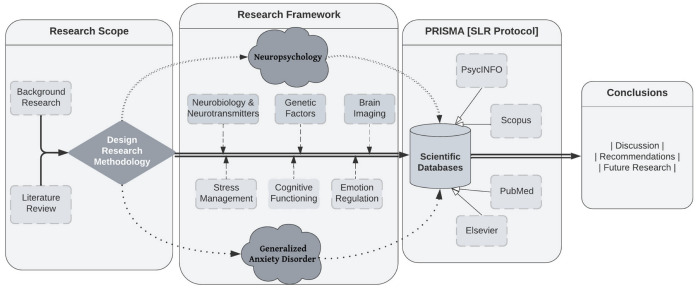
Flowchart of research framework.

**Figure 2 healthcare-11-02446-f002:**
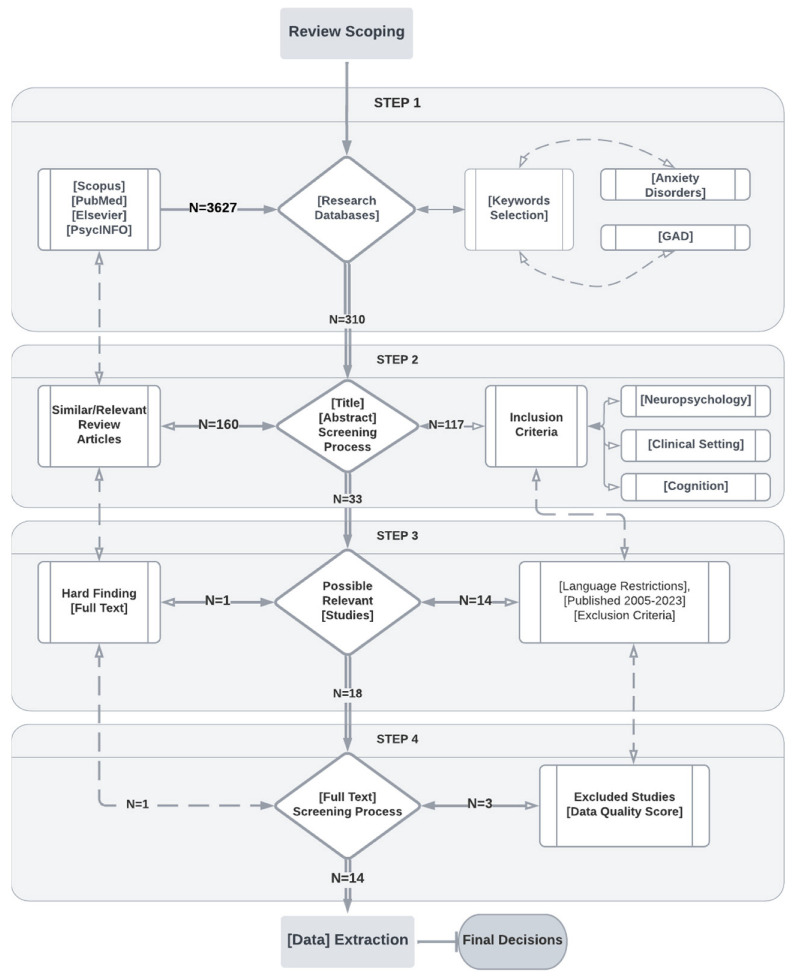
Flowchart of the PRISMA methodology.

**Table 1 healthcare-11-02446-t001:** Empirical studies (author, sample, measures, results).

Authors	Sample	Measures	Results
Aasvik et al. (2015) [23]	167	EMQ-R, SF-8, HADS, CFQ, ISI	Significant levels of fatigue and anxiety were reported while depression, insomnia, and intensity of pain levels were not significant Subjective memory complaints may reflect concerns about one’s own memory performance, so they are expressions of anxiety, becoming anxious when asked to remember something Anxiety consumes attention
Balderston et al. (2017) [26]	69	BAI, STAI, BDI, WASI	Anxiety patients have slower reaction times, difficulty recruiting some regions in cognitive tasks, intrusive thoughts and low control over them, low emotion regulation, and continuous threat-related thoughts during tasks, and they must work to face fearful stimuli
Fitzgerald et al. (2017) [27]	69	ERT	Trait anxiety patients have dIPFC-mediated cognitive control abnormalities WM deficiencies may predict future illness severity or treatment outcomes
Fonzo et al. (2014) [28]	32	PSWQ, 10 sessions of weekly CBT, Emotion Face Assessment Task	The results support a dual-process psychotherapy model of neural system modifications in generalized anxiety disorder that attenuates cingulo-amygdala responsiveness to threat signals and potentiates insular responses to pleasant facial emotions
Gordeev et al. (2013) [10]	95	Clinical–neurological, neuropsychological methods	Generalized anxiety disorder patients had more anxiety, depression, and short-term memory and directed attention disorders than panic disorder patients In panic patients, P300 amplitudes were higher
Hallion et al. (2017) [11]	56	MINI, CSR, CGI, SIGH-A, PSWQ, SIGH-D	Generalized anxiety predicts impaired “cool” and cognitive inhibition but not worry Anxiety affects cognitive efficiency by requiring more effort (reflected in part by slower response times) to maintain accuracy (reflected in task accuracy), which is partly due to worry competing for attentional resources
Khdour et al. (2016) [12]	73	NAART, WAIS-R, Digit Span test, HAM-A	Negative feedback helped generalized anxiety patients. Cognitive dissociation between anxiety spectrum disorder subtypes may explain differences in neural circuitry People with generalized anxiety learn better from negative feedback but not because of group differences in learning speed or ability to explore outcomes
Leonard and Abramovitch (2018) [1]	1563	MINI, PSWQ, DASS-21, STAI, NeuroTrax Computerized Neuropsychological Battery	No significant differences were found for any neuropsychological outcome measures or domain indexes
Moon et al. (2015) [13]	36	HAMD 17, GAD-7, STAI-I, STAIII, ASI-R	Generalized anxiety disorder patients performed significantly worse on all questionnaires than healthy controls Generalized anxiety patients reacted to anxiety-related situations with fear, lower accuracy, cognitive deficits, and low attention
Plana et al. (2014) [29]	2738	40 studies evaluating mentalization, emotion, social perception/knowledge, or attributional style in anxiety disorders	Post-traumatic stress disorder patients had mentalization and emotion recognition deficits, while other anxiety disorders showed attributional biases
Renna et al. (2018) [30]	17	Structured clinical interview Anxiety disorders interview schedule	Generalized anxiety disorder patients have attention issues Generalized anxiety is characterized by worry, which prevents distraction and draws attention to the threat. Lack of attention reduces social skills
Stefanopoulou et al. (2014) [14]	17	Penn State Worry Questionnaire, BDI-II, N-Back Task, Random Generation Key-Pressing Task, mood ratings, WTAR	Generalized anxiety disorder patients have fewer attentional control resources while worrying Anxious people have fewer resources to perform concurrent thinking tasks when thinking about personal topics Verbal preoccupation requires less attentional control, suggesting negative biases use resources
Tempesta et al. (2013) [31]	40	STAI, BDI, PSQI, TAS-20	The WCST showed that young subjects’ executive functions and immediate recall were affected Antidepressants reduced sustained attention performance, making it harder to stay alert
White et al. (2017) [32]	78	A passive avoidance task	Generalized anxiety disorder impaired reinforcement-based decision making

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
