# Peer review of "Neuropsychology of Generalized Anxiety Disorder in Clinical Setting: A Systematic Evaluation"

_healthcare, 2023, doi:10.3390/healthcare11172446_

Round 1

Reviewer 1 Report

The article tries to contribute to the field of Neuropsychology (NPs) and GAD; the method used is systematic review.

It is a good idea, and a really great effort, but the actual status of the manuscript is perfectible.

General: number of references justifying the content is insufficient, there are many paragraphs without references.

Abstract: does not indicate the justification and utilitity of the study; one of the objective is not really important (publication and volume); and it does not refer about conclusiones or general results.

Introduction: too broad and not NPs oriented. The form to presente objective seems inadecuate; and the objective 1 is bibliométric; something unnecessary in comparison with the metdodological and clinical objectives. Think about the elimination of the RQ1

Literature Review:  somewhat unsubstantiated, as in general the manuscript. Very general, extensive in terms of the description of the GAD but not oriented to the focus of the article.

Method: It is important to improve the method, in-exclusion criteria and procedures. It is not clear what data and how 2 researchers can examine the documents 311,672.

The selection, eligibility and inclusion procedure are unclear.

There no information about the results of Scopus, Pub Med, or Elsevier... only of PsycInfo. The results of only 13 studies who use NPs assesstment is surprising and the procedimmment to reach it is unclear

The figure 1 is unncesary

Results

Table 1;  the Type of study column is constant, so unnecessary; the conlcusión of studies are not speccifically oriented to NPs, none of the instrument are cited.

Many of the contents of resuls writed are not part of the incluided articles.

Is important a synthesis of the evidence, or results found, but of the findings.

Dicussion:

does not discuss background with the findings, but provides further information, weaknesses, future directions and practical usefulness are not made clear

Formal: review the redaction of p.4 line 175 "since I have..."

Author Response

The paper has undergone significant enhancement due to your comments, particularly concerning the suggested modifications on the following topics. The references have been enhanced with additional studies, the abstract has been revised following the provided feedback and concerns, the introduction has been aligned with the field of neuropsychology, and the original research hypothesis 1 has been excluded. The literature review underwent a comprehensive revision in alignment with the research objectives. The numbers in the method were revised as a result of random mistakes. At the same time, a detailed description of the studies was provided in Table 1, and the associated text in the results section.
At the end of the discussion, a comprehensive examination of the findings was conducted, incorporating the suggested sections, namely weaknesses and potential avenues for further research. The comments and directions provided were crucial in enhancing the quality of this manuscript.

Reviewer 2 Report

Interesting work in the sense that it attempts to analyze the different contributions regarding a specific topic, which can be helpful in establishing the essence of Neuropsychology and GAD. In fact, its explicit goal is to validate the primary literature from the last ten years that addresses neuropsychological functioning in various cognitive domains in individuals with generalized anxiety disorder. However, later in the development of the work, they break or exceed this framework.

there is a gap in the sequence of references, specifically, references 3, 4, and 5 are missing between references 2 and 6.

Introduction and discussion should be better connected. In the Introduction, including the "Literature Review" section, the problem should be presented, and in the discussion, the results should be analyzed based on what was previously stated.

The "Literature Review" section consists of an exhaustive list of topics and lacks proper structure. Figure 1 helps in understanding it, although it could also be improved. The structure is somewhat confusing with an inappropriate use (in my opinion) of bold highlights. As mentioned earlier, Figure 1 aids in organizing the previous information, providing a coherent order to the list of topics.

Several databases are mentioned in the text and Figure 1, but in the Materials and Methods section, line 169, it is stated that only PsycINFO was used. The use of these specific databases should be clarified and justified. Why these databases and not others? Additionally, the years used should also be justified. For instance, including the year 23 seems odd when there is still a considerable amount of time left for it to finish.

The descriptors used for the search should also be justified. In the same section, line 175, there is a statement: "since I have focused my systematic review on that," which appears to be an error and needs to be corrected or clarified, as the work is signed by two.

Section 4.1 needs to be detailed more clearly. It wouldn't hurt to clarify the different steps. For example, they mention starting with 33 articles, selecting 18 for full-text evaluation, eliminating 5 after this evaluation, and ending up with 13... It should be made clearer how the different articles are filtered throughout this process.

Table 1 should be improved as there is too much text in some columns, making it complicated to read. Additionally, the "Type of Study" column could be removed since all the works are quantitative. The "Sample" column should be more systematic. For instance, the first work mentions 167 patients, the second mentions "69 participants from the Washington DC metropolitan area." It would be better to provide location information for the first or ignore it for both cases, but in a consistent manner. Similarly, in the next column "Instrument," there are inconsistencies - the first case mentions the instruments used, while the second one states that "Participants completed measures of anxiety." The information in the table should be well-structured and coded.

As the table occupies more than one page, the first row should be made into a header to facilitate the optimal flow of information.

The references from the search, 18, 19, 20, 22, 23, and 24, are not referenced in the Results section. As it is a systematic review, the contribution of these works should be included, even if it is in relation to the others. On the other hand, different topics or descriptors are introduced, supporting references unrelated to the previous list.

Author Response

The comments provided have significantly enhanced the quality of the work, particularly concerning the subsequent focus areas. The manuscript has been updated to include additional references, the abstract has been edited based on the feedback and concerns raised, the introduction has been refocused to emphasize the field of neuropsychology, and the original research hypothesis 1 has been eliminated. Following the research objectives, the literature review was comprehensively revised. As a result of unintentional errors, the numerical values presented in the technique section have been changed. Furthermore, in the results section, Table 1 and the corresponding textual description now provide a focused and specific account of the conducted investigations. Your comments and suggestions were crucial and decisive in enhancing the quality of this paper.

Round 2

Reviewer 2 Report

The new version is a clear improvement with respect to the first manuscript. The introduction is presented more clearly and developed, although I do not finish seeing the usefulness to separate introduction and literature review, both of them suppose the introduction.

Made of less an answer for my comments, since the answer they provide is too general.

Some issues that have not been too well resolved are:

-They state that their research covers the last ten years (2005-2023). As far as I know, from 2005 to 2023 is 18 years, and I still think that including 2023, with almost half a year left in the year, isn't quite accurate. Additionally, upon reviewing the analyzed contributions, these range from 2013 to 2018.

-I insist that the process should be clarified further. They retrieve 117 articles, narrow it down to 33, from which they select 18 for a full-text assessment, and ultimately end up with 14 (one more than in the initial version). It's worth expanding a bit more to explain these steps.

-I understand that they attempted to improve Table 1, but now it's almost worse, at least in terms of formatting. Since it spans multiple pages, it should have a header row. It's worth putting more effort into refining the visual format, reducing the number of divided cells.

I've reviewed the listing in Table 1 and I still can't grasp the order of contributions, as it's neither chronological nor alphabetical.

-I continue to feel that the various contributions listed in Table 1 should be systematically integrated into the results and discussion, emphasizing or highlighting the different insights that can be drawn from their analysis. On the contrary, they overuse the description of topics as a list (they also do this in the Literature Review section), where in many cases there isn't even a supporting reference. From line 399 to 422, there isn't a single reference, yet they describe 5 topics.

-Lastly, the Introduction (Introduction and Literature Review) should be better connected. In the discussion, they use 24 different references, and only 8 have been mentioned previously in the introduction. The introduction must clearly reflect the framework's setup, making it an important part of the discussion.

Author Response

We would like to express our gratitude to the reviewers for their insightful feedback pertaining to enhancing the quality of our research. Consequently, we would like to acknowledge the modifications we implemented in response to the comments provided by the reviewers. The research incorporated in this study covers from 2012 to 2022. In the methods section, further details were included regarding the process of picking articles based on study hypotheses. Specifically, attention was placed on studies that emphasized neurocognitive factors. Table 1 underwent a comprehensive reformulation in accordance with the recommendations put forth by the panel of reviewers, resulting in a notable enhancement of its visual presentation.

The integration and cohesion of the introduction and literature review sections were enhanced. Simultaneously, as properly noted, citations from lines 399-422 were incorporated.